# Oncolytic adeno-immunotherapy modulates the immune system enabling CAR T-cells to cure pancreatic tumors

Amanda Rosewell Shaw [1,2,4], Caroline E. Porter[1,2,4], Tiffany Yip[1,2], Way-Champ Mah[1,2], Mary K. McKenna[2,3], Matthew Dysthe[2,3], Youngrock Jung[1,2], Robin Parihar[2,3], Malcolm K. Brenner[1,2,3] & Masataka Suzuki [1,2✉]

High expression levels of human epidermal growth factor receptor 2 (HER2) have been associated with poor prognosis in patients with pancreatic adenocarcinoma (PDAC). However, HER2-targeting immunotherapies have been unsuccessful to date. Here we increase the breadth, potency, and duration of anti-PDAC HER2-specific CAR T-cell (HER2.CART) activity with an oncolytic adeno-immunotherapy that produces cytokine, immune checkpoint blockade, and a safety switch (CAdTrio). Combination treatment with CAdTrio and HER2.CARTs cured tumors in two PDAC xenograft models and produced durable tumor responses in humanized mice. Modifications to the tumor immune microenvironment contributed to the antitumor activity of our combination immunotherapy, as intratumoral CAdTrio treatment induced chemotaxis to enable HER2.CART migration to the tumor site. Using an advanced PDAC model in humanized mice, we found that local CAdTrio treatment of primary tumor stimulated systemic host immune responses that repolarized distant tumor microenvironments, improving HER2.CART anti-tumor activity. Overall, our data demonstrate that CAdTrio and HER2.CARTs provide complementary activities to eradicate metastatic PDAC and may represent a promising co-operative therapy for PDAC patients.

[1] Department of Medicine, Section of Hematology/Oncology, Baylor College of Medicine, Houston, TX, USA. [2] Center for Cell and Gene Therapy, Baylor College of Medicine, Texas Children's Hospital, Houston Methodist Hospital, Houston, TX, USA. [3] Department of Pediatrics, Section of Hematology/Oncology, Baylor College of Medicine, Houston, TX, USA. [4] These authors contributed equally: Amanda, Rosewell Shaw, Caroline E, Porter. ✉email: suzuki@bcm.edu

Pancreatic cancer is the seventh leading cause of cancer death worldwide and without new effective treatments will soon surpass the death rate of more common cancers[1]. Pancreatic ductal adenocarcinoma (PDAC) tumors tend to have low mutational burden[2] and are classified as immunologically "cold" tumors with little immune cell activation within the immunosuppressive tumor microenvironment (TME)[3]. Thus, even in the era of cancer immunotherapy, PDAC patients rarely benefit from immunotherapies like checkpoint blockade[3].

Oncolytic viruses (OVs) selectively replicate in and kill tumor cells and can induce immune cell infiltration at tumor sites through virus-mediated inflammatory responses[4]. Based on this unique feature, OVs have been investigated in PDAC patients to overcome limited immune infiltration into tumors. Talimogene laherparepvec (T-VEC), an oncolytic herpesvirus FDA-approved for the treatment of melanoma, showed some anti-tumor activity in patients with advanced pancreatic cancer; however, the study was terminated early due to the rate of patient progression[5]. Likewise, pelareorep, an oncolytic reovirus, tested in a phase II trial demonstrated safety but did not improve progression-free survival[6]. On the other hand chimeric antigen receptor-modified T cells (CAR T cells) are gaining attention as potential therapeutic options for PDAC patients. Although Mesothelin-specific CAR T cells demonstrated safety, they had limited anti-tumor activity in PDAC patients[7]. These clinical results indicate that single immunotherapy agents are insufficient to control PDAC tumor growth, and recent clinical trials for PDAC have combined different types of agents (e.g., combination of radiotherapy, chemotherapy, vaccine plus checkpoint inhibitors (NCT02648282)) to overcome multiple barriers to immunotherapy.

High HER2 expression is associated with poorer prognosis in PDAC patients[8]. In a phase II clinical trial, patients with metastatic PDAC with grade 3 HER2 expression by immunohistochemistry were treated with the HER2 monoclonal antibody Trastuzumab (Herceptin) in combination with fluorouracil[9]. This HER2-targeting therapy was tolerated but did not improve progression-free or overall survival in these patients.

We previously demonstrated that our combination immunotherapy strategy, which couples oncolytic adenoviral immunotherapy (CAd)[10] with clinically tested HER2-specific CAR T-cells (HER2.CART)[11,12], mediates significant anti-tumor effects against multiple solid tumors[13]. Our combination of CAd expressing IL-12 and PD-L1 blocking antibody with HER2.CART improved long-term survival in an aggressive metastatic model of head and neck squamous cell carcinoma[14,15]. Here, we tested whether our combination immunotherapy with HER2.CART and CAd expressing IL-12, PD-L1 blocking antibody, and safety switch HSVtk (CAdTrio) could resolve HER2-positive PDAC tumors.

We found that one clinically feasible dose of CAdTrio was unable to control tumor growth in PDAC xenograft mouse models. However, even local CAdTrio treatment stimulated systemic host immune responses to repolarize distant PDAC tumor microenvironments, additively improving HER2.CART anti-tumor activity to eradicate advanced PDAC tumors in humanized mice.

## Results

**HER2.CARTs and CAdTrio kill human PDAC lines in vitro.** Although monoclonal antibody therapy targeting HER2 previously failed for PDAC patients[9], HER2.CARTs have higher avidity (multivalent versus bivalent) and ability to directly kill tumor cells through cytolytic proteins as opposed to antibody-dependent cellular cytotoxicity through infiltrating immune cells. We thus hypothesized that our clinically tested HER2.CARTs[11]

could efficiently target HER2-positive PDAC tumors. We first confirmed that our HER2.CARTs effectively kill PDAC lines with varying levels of HER2 expression (Fig. 1a). Since PDAC patients have been safely treated with oncolytic viruses[16,17], we also confirmed that our oncolytic adeno-immunotherapy, comprised of an oncolytic adenovirus (OAd) and a helper-dependent adenoviral vector (HDAd) encoding human interleukin 12p70 (hIL12p70), programmed death ligand 1 (PDL1) blocking mini-antibody, and herpes simplex virus thymidine kinase (HSVtk) safety switch expression cassettes (CAdTrio), lysed PDAC lines (Fig. 1b). We next assessed whether CAdTrio-infected PDAC lines amplified the transgenes encoded in HDAd (Fig. 1c, Supplementary Fig. 1). We found that CAdTrio induced oncolysis of PDAC lines at levels similar to OAd alone (Fig. 1b), and transgenes encoded in HDAd were significantly amplified in CAdTrio-infected PDAC cells compared to those infected with only HDAd (Fig. 1c). To test whether these transgenes improved HER2.CART killing, we co-cultured PDAC cells infected with HDAdTrio (no lytic effect) with HER2.CART (Supplementary Fig. 2). We confirmed that HDAd-derived transgenes enhanced HER2.CART killing of PDAC cells in vitro, similar to other tumor types, even in the absence of adenoviral mediated oncolysis[13,14].

**HER2.CARTs are the primary mediator of PDAC tumor control in xenograft models.** To evaluate the complementary benefits of CAdTrio and HER2.CARTs in vivo, mice with subcutaneous CFPAC-1 tumors received $1 \times 10^7$ viral particles (vp) of CAdTrio before being infused with $1 \times 10^6$ HER2.CARTs 3 days later (Fig. 2a). Although CAdTrio failed to control tumor growth, mice treated with either HER2.CART alone or combination of HER2.CART and CAdTrio both had consistent, complete, and sustained eradication of CFPAC-1 tumors (complete response: CR). In these CFPAC-1 mice we observed no difference in HER2.CART infiltration or expansion at the tumor site or circulating IFNγ levels in the blood of mice treated with either HER2.CART alone or combination therapy (Fig. 2b, c). Likewise, there was no difference in animal survival between HER2.CART alone and combination treatment (Fig. 2d). Although we detected CAdTrio-derived IL-12p70 in the blood (Fig. 2c), we found local CAd treatment had no additive anti-tumor effect in CFPAC-1 xenograft mice.

To address whether the observed anti-tumor effects were unique to CFPAC-1, we evaluated the reproducibility of our results in a subcutaneous model of CAPAN-1 (Fig. 3a). CAdTrio alone showed little antitumor activity and we observed no statistical difference in survival between control and CAdTrio-treated mice (Fig. 3d). Similar to a previous study, we still detected Ad vectors in tumors (Supplementary Fig. 3a)[10]. Thus, CAdTrio is likely unable to control tumor growth in these xenograft models due to necrosis and disruption of tumor vasculature by oncolysis, which may prevent the continuous spread of viruses within the tumor mass[10].

HER2.CART alone significantly controlled tumors compared to the control group. However, tumors recurred in two HER2.CART-treated mice due to antigen (HER2) loss (Supplementary Fig. 3b). We found, however, that combination treatment with CAdTrio and HER2.CART completely and consistently eradicated CAPAN-1 tumors without tumor relapse (CR). We also found that mice pre-treated with CAdTrio showed significantly higher HER2.CART infiltration/expansion ($p < 0.005$) at the tumor site within 7 days post infusion and increased circulating IFNγ ($p = 0.004$) in the blood compared to mice treated with HER2.CART alone at 7 days post infusion (Fig. 3b, c). These results indicate that oncolysis combined with PD-L1 blockade and IL-12 enhances the potency of adoptively transferred HER2.

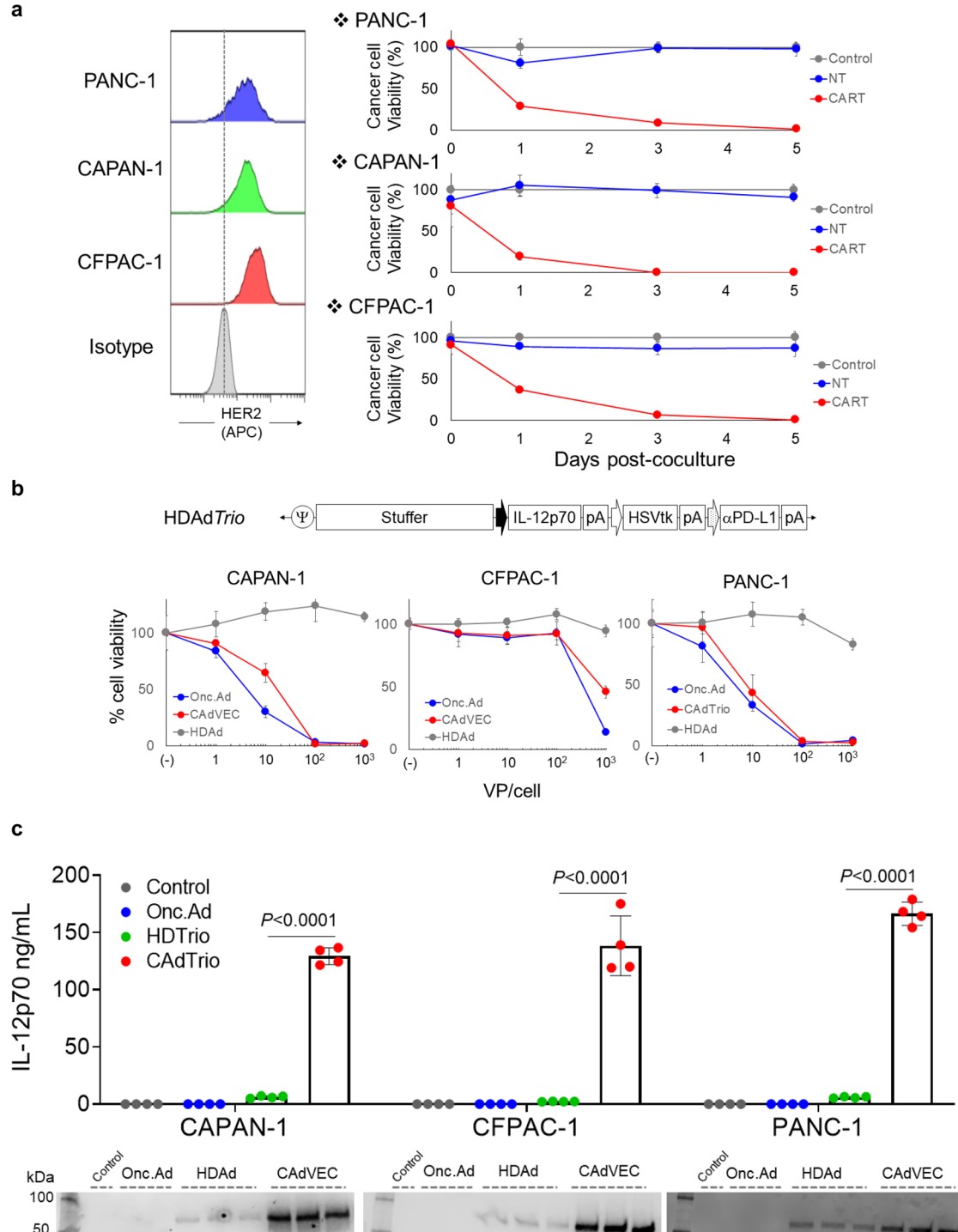

**Fig. 1 HER2.CART and CAdTrio kill human PDAC lines in vitro. a** HER2 expression was analyzed by flow cytometry on PANC-1, CAPAN-1, and CFPAC-1. Tumor cells expressing *ffLuc* were cultured with NTs or HER2.CARTs (E:T = 1:10). Cells were harvested 0, 1, and 5 days post coculture, and viable cancer cells were analyzed by luciferase assay ($n = 4$ biologically independent samples, each time point). Data are presented as means ± SD. **b** Schematic structure of HDAd encoding human IL-12p70, HSVtk safety switch and PDL1 blocking antibody expression cassettes (HDAd*Trio*). PANC-1, CAPAN-1, and CFPAC-1 were infected with increasing doses of HDAd*Trio*, OAd, or CAd*Trio* (OAd: HDAd=1:1) ($n = 6$ biologically independent samples). We analyzed viable cells at 96 h by MTS assay. Data are presented as means ± SD. **c** PANC-1, CAPAN-1, and CFPAC-1 were infected with total 10 vp/cell of HDAd*Trio*, OAd, or CAd*Trio* (OAd:HDAd=1:1) ($n = 4$ biologically independent samples). We sampled media 48 h post infection and quantified levels of IL-12p70 and PD-L1 mini-antibody by IL-12p70 ELISA assay and Western blotting for PD-L1 mini-antibody, respectively. Data are presented as means ± SD, $p < 0.0001$ determined by two-tailed $t$ test ($t = 32.93$, dF = 6). Statistical significance set at $p < 0.05$, ns $> 0.05$.

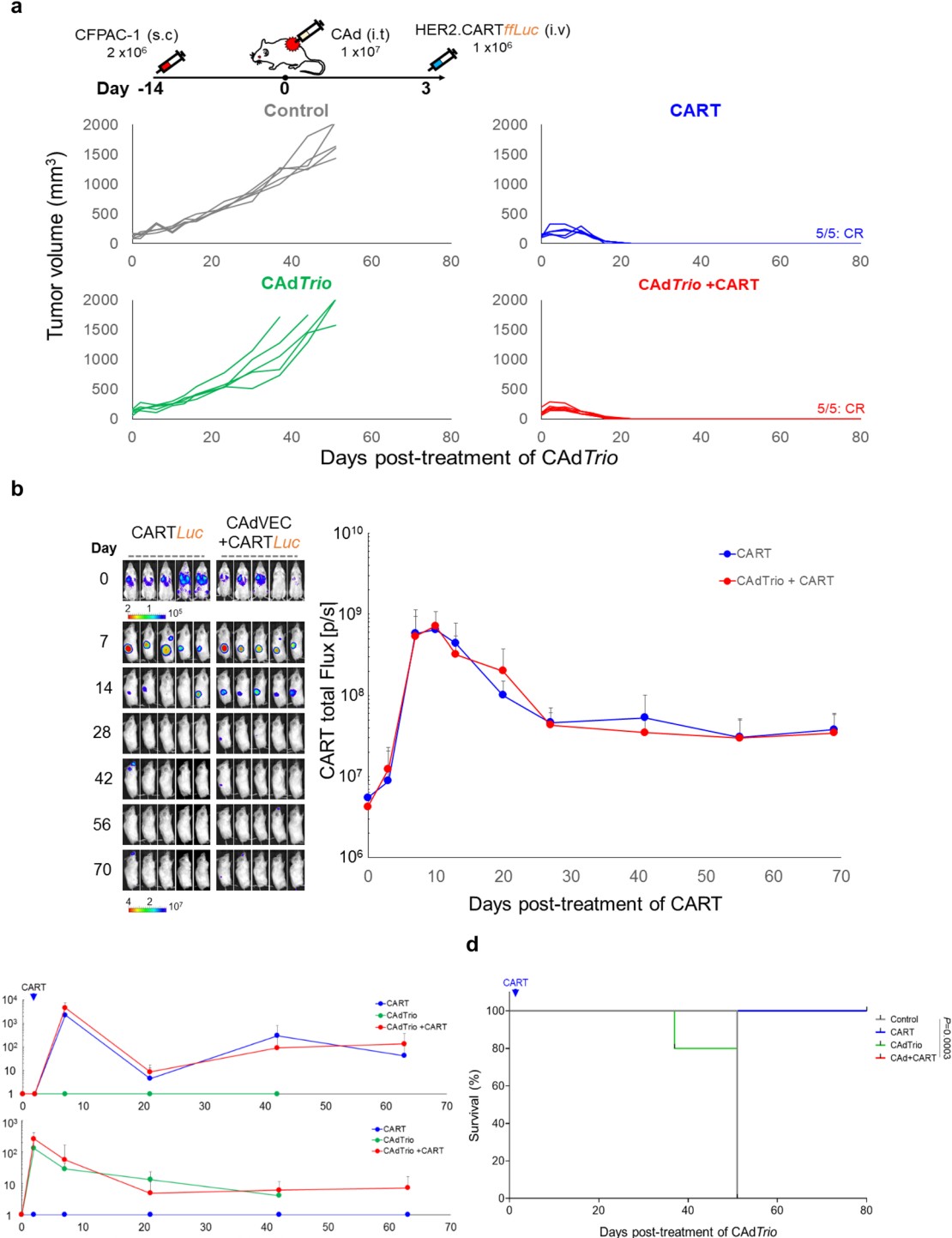

**Fig. 2 HER2.CARTs primarily control CFPAC-1 tumor growth in xenograft mouse model. a** CFPAC-1 cells were transplanted into the right flank of NSG mice ($n = 5$ animals). A total of $1 \times 10^7$ vp of *CAdTrio* (OAd:HD = 1:1) were injected i.t.. A total of $1 \times 10^6$ HER2.CARTs expressing *ffLuc* were administered i.v. 3 days post injection of *CAdTrio*. Tumor volumes were monitored at different time points. **b** Bioluminescence of HER2.CARTs was monitored at the indicated time points. Data are presented as means ± SD. **c** We collected serum samples from mice at 0, 3, 7, 21, 42, and 63 days post injection of *CAdTrio*, and measured IFNγ and IL-12p70 levels by ELISA. Data are presented as means ± SD. **d** Kaplan–Meier survival curve after *CAdTrio* administration in mice ($n = 5$ animals) $p = 0.0003$. *P*-values were determined using the log-rank Mantel–Cox test (dF=3). Statistical significance set at $p < 0.05$, ns > 0.05. Abbreviations: s.c. subcutaneous, i.t. intratumoral, i.v. intravenous.

CARTs to control tumor growth in subcutaneous CAPAN-1 tumors. In this model, however, HER2.CART remained the major contributor to CAPAN-1 tumor control, as there was no significant difference in survival between HER2.CART alone and combination treatment (Fig. 3d). These results indicate that

HER2.CARTs effectively recognize and control PDAC tumor growth in vivo, and the additive effect of *CAdTrio* depends on the PDAC line. We did not observe autonomous HER2.CART expansion-related toxicity (e.g., weight loss) in these xenograft models (Supplementary Fig. 4).

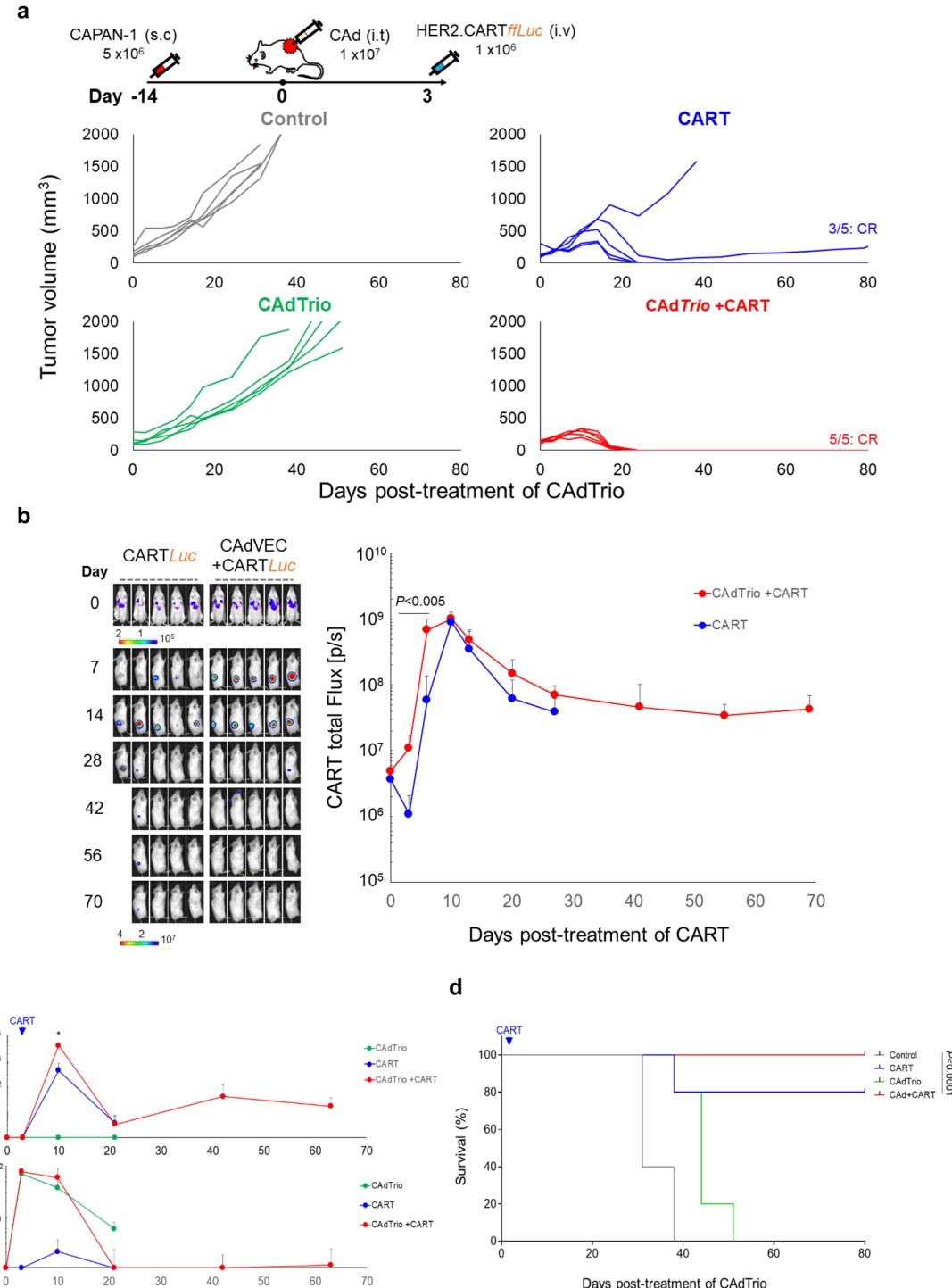

**Fig. 3 HER2.CARTs primarily control CAPAN-1 tumor growth but require CAd*Trio* to cure PDAC in xenograft mouse model. a** CAPAN-1 cells were transplanted into the right flank of NSG mice ($n = 5$ animals). A total of $1 \times 10^7$ vp of *CAdTrio* (OAd:HD = 1:1) were injected into the tumor. A total of $1 \times 10^6$ HER2.CARTs expressing *ffLuc* were systemically administered 3 days post injection of *CAdTrio*. We monitored tumor volumes at different time points. **b** We monitored HER2.CART bioluminescence at different time points. Data are presented as means ± SD, $p < 0.005$. *P*-values were determined by two-tailed *t* test ($t = 4.32$, dF = 8). **c** We collected serum samples from mice at 0, 3, 10, 24, 45, and 66 days post injection of *CAdTrio*, and measured levels of IFNγ and IL-12p70 by ELISA. Data are presented as means ± SD, $p = 0.004$. *P*-values were determined by two-tailed *t* test ($t = 5.769$, dF = 8). **d** Kaplan–Meier survival curve after *CAdTrio* administration in mice ($n = 5$ animals), $p < 0.0001$. *P*-values were determined using the log-rank Mantel–Cox test (dF = 3). Statistical significance set at $p < 0.05$, ns > 0.05. Abbreviations: s.c. subcutaneous, i.t. intratumoral, i.v. intravenous.

**Combination treatment is required to effectively control PDAC in humanized mice.** The tumor microenvironment (TME) contributes to immunotherapy resistance in solid tumors, including PDAC[18]. To address whether our combination immunotherapy is required to overcome the protective TME, we evaluated its antitumor activity in immunocompetent animal models. Since human Ad-based OADs have limited infectivity (our Ads: serotype 3 knob) and replication in rodent cell lines[19,20], and our HER2.CAR construct was optimized for human T cells[21], we reconstituted transgenic NSG mice ubiquitously expressing human SCF, IL-3, and GM-CSF (NSG*SGM3*) with human innate and adaptive immune cells using HLA-A2 cord blood unit (CBU)-derived CD34$^+$ cells (Supplementary Fig. 5a)[22]. We then transplanted PDAC tumors into these humanized mice using CFPAC-1, which is HLA-A2 (Supplementary Fig. 5b). We found minimal systemic inflammation due to CFPAC-1 tumor formation with tumor-infiltrating human immune cells (Supplementary Fig. 5b, c), suggesting that these humanized mice tolerated CFPAC-1. To avoid allo-reaction to adoptively transferred HER2.CARTs, we generated HER2.CART from the same CBU-derived T cells (Supplementary Fig. 6a). These CBU-derived HER2.CARTs had phenotypes and CFPAC-1 killing similar to HER2.CART generated from healthy donor-derived PBMCs (Supplementary Fig. 6a).

After CFPAC-1 tumor formation, we treated mice with the same conditions as in the xenograft models: $1 \times 10^7$ vp CAd*Trio* and $1 \times 10^6$ autologous HER2.CARTs infused three days later (Fig. 4a). Although CAd*Trio* had no effect on CFPAC-1 tumors in our xenograft model (Fig. 2), in the humanized model CAd*Trio* alone led to significant ($p = 0.0002$) tumor control and prolonged survival ($p = 0.0107$) compared to the untreated control group (Supplementary Figs. 7 and 8). These results suggest that the host immune system, stimulated by CAd*Trio*, contributes to tumor control that cannot be seen in immunodeficient xenograft models where tumor control is primarily dependent on oncolysis. In contrast to the CFPAC-1 xenograft model (Fig. 2), HER2.CART alone delayed tumor control but did not cure all treated animals (Fig. 4a). In this humanized CFPAC-1 model, combination therapy had additive anti-tumor effects, with most treated mice achieving CR. There was no significant induction of pro-inflammatory cytokines (IL-6, TNFα, IL-1β) or Th1-related cytokines (IFNγ, IL-2, IL-7) in the serum of humanized mice treated with single agents or combination treatment at early time points (Fig. 4b, Supplementary Fig. 9), indicating that these doses are well tolerated.

Due to lack of appropriate inflammatory and chemokine signals, CARTs show limited trafficking and infiltration to tumor sites in patients with solid tumors[23,24]. We found that combination treatment significantly increased early HER2.CART infiltration to the tumor site ($p = 0.003$) compared to HER2.CART alone in humanized mice (Fig. 4c). Oncolytic virotherapy stimulates the host immune system through pattern recognition receptors and induces pro-inflammatory cytokine and chemokine expression[25,26]. To address whether enhanced infiltration of HER2.CART was dependent on chemotaxis induced by CAd*Trio*, we harvested tumors 3 days post injection of CAd*Trio* (time at which CART would have been infused) and quantified pro-inflammatory RNA expression at the tumor site (Fig. 4d). Although CAd*Trio* did not induce pro-inflammatory genes in CFPAC-1 tumors in xenograft mice (lacking immune cells), in humanized mice, CAd*Trio* induced expression of type I IFN, IFNβ ($p = 0.0093$), and type I IFN-dependent genes, including OAS1 ($p = 0.0004$), Mx1 ($p = 0.0005$), CCL5 (RANTES), and CXCL10 (IP10, $p = 0.0045$)[27–30]. Since there was no significant difference in infiltrating immune subsets between untreated control and CAd*Trio*-treated humanized mice (Fig. 4e), these

results suggest that tumor-infiltrating immune cells recognize CAd*Trio* infection[29,31] and stimulate the local immune system through type I IFN[29]. This response also leads to expression of type II IFN, IFNγ, and IFNγ downstream genes (PD-L1, ICAM-1) at the tumor site (Fig. 4d)[13]. We confirmed that CBU-derived HER2.CARTs highly express chemokine receptors, CCR5 (for CCL5) and CXCR3 (for CXCL10) (Supplementary Fig. 6b). These results indicate that local CAd*Trio* treatment induces pro-inflammatory signals from the TME (but not CFPAC-1 cells themselves) and increases infiltration of adoptively transferred HER2.CART at the pre-treated tumor site, similar to a previous report[32].

There was no autonomous expansion of adoptively transferred HER2.CART or induction of pro-inflammatory/Th1 cytokines in serum of humanized mice treated with combination immunotherapy, suggesting that CAd*Trio* locally enhances HER2.CART anti-tumor effects, leading to superior anti-tumor effects (Figs. 4a, 4 of 6 mice: CR) without systemic toxicity.

**Combination treatment controls both CAd*Trio*-treated and untreated PDAC tumors in humanized mice.** Results from our humanized mouse model (Fig. 4) indicate that CAd*Trio* stimulates the host immune system and modulates the TME to recruit adoptively transferred HER2.CART to the tumor site. We thus hypothesized that CAd*Trio*-activation of the host immune system also modulates the TME at a distant tumor site (abscopal effect), leading distant tumor beds to respond to adoptively transferred HER2.CART. Since metastasis correlates with poor prognosis in PDAC patients[33], we gave our humanized mice a large (primary: right) and a small (metastatic: left) tumor to mimic metastatic disease, and treated only the right tumor with CAd*Trio* (Fig. 5a). Although both tumor sites in mice treated with HER2.CART alone showed HER2.CART infiltration 3 days post infusion (Fig. 5b), CART had limited expansion compared to the single tumor model, resulting in no significant tumor control compared to control mice (Supplementary Fig. 10). These results suggest that a threshold of CART infiltration is required to control CFPAC-1 tumor growth in humanized mice.

Local CAd*Trio* treatment controlled growth of the right (treated) tumor similar to our single tumor model but did not significantly control the left (untreated) tumor (Fig. 5a). However, even in this advanced tumor model, combination treatment completely eliminated right (CAd*Trio*-treated) tumors and significantly controlled distant tumor growth compared to other groups ($P < 0.01$) (Fig. 5a, Supplementary Fig. 10). Based on the CAd*Trio*-dependent chemotaxis observed in our single tumor model, we investigated HER2.CART infiltration at CAd*Trio*-treated tumor site (Fig. 5b). Combination treatment increased HER2.CART infiltration more at the right tumor than at the left at 3 days post infusion, but HER2.CART expanded at both sites equally after 14 days. Thus, improved HER2.CART infiltration and expansion leads to sustained anti-tumor effects even in CAd*Trio*-untreated tumors.

To address which CAd*Trio* components (oncolysis, IL-12 and PD-L1 blocking antibody) contribute to sustained anti-tumor effects with HER2.CART in humanized mice, we administered CAd0 (no transgene), CAd*PDL1* (PD-L1 blocking antibody), or CAd*IL12* (IL-12p70) (Supplementary Fig. 15) to the right tumor of humanized mice bearing two subcutaneous CFPAC-1 tumors, as in the previous experiment. Although each CAd component (oncolysis, PD-L1 blocking antibody or IL-12) provided some level of tumor control in both CAd-treated and untreated tumors when combined with HER2.CART, none eliminated the CAd-treated tumor. These results suggest that all components are required to control primary (large) tumor growth

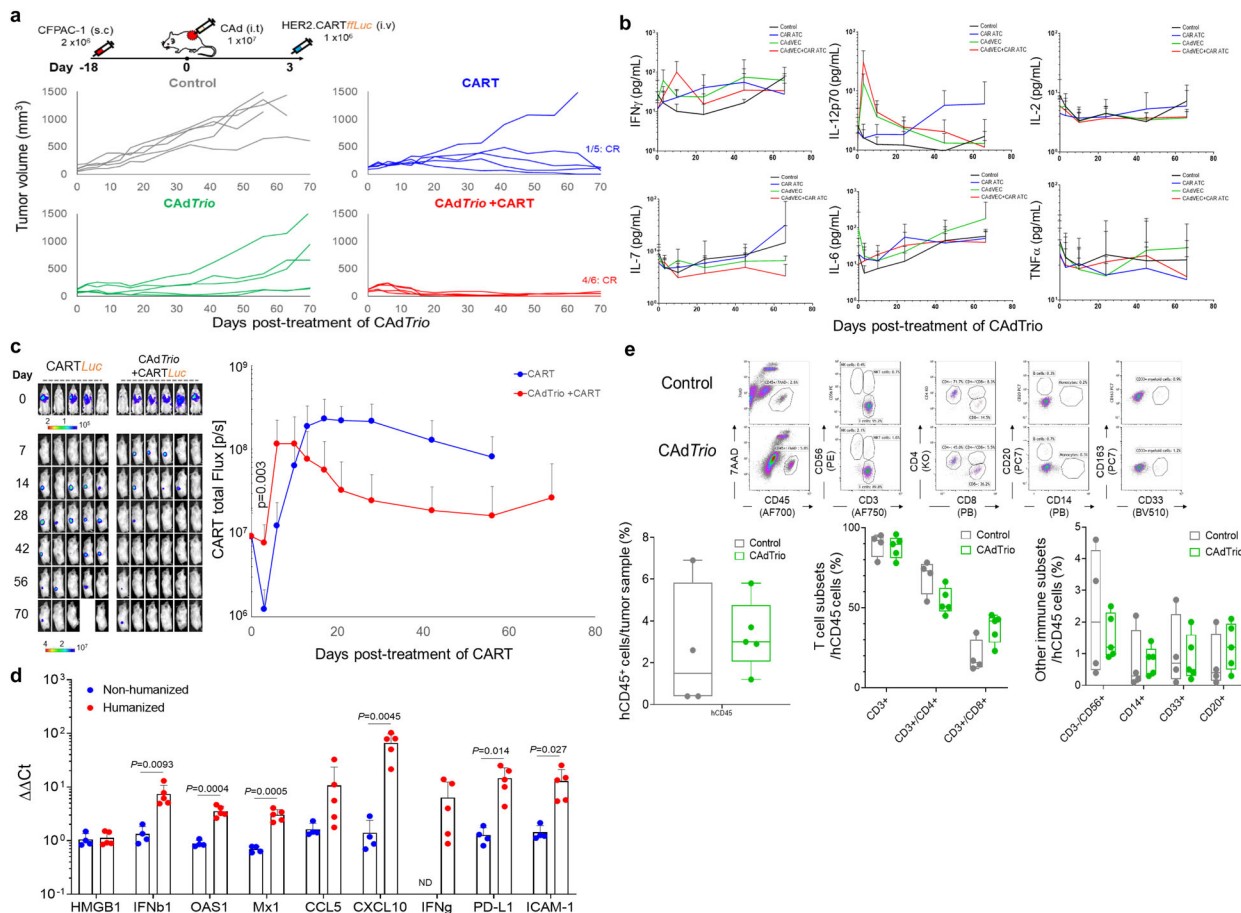

**Fig. 4 Combination immunotherapy controls CFPAC-1 tumor growth in humanized mouse model. a** CFPAC-1 cells were transplanted into the right flank of humanized mice (control, CART alone, CAd*Trio* alone: $n = 5$ animals, CAd*Trio*+CART: $n = 6$ animals). A total of $1 \times 10^7$ vp of *CAdTrio* (OAd:HD = 1:1) were injected into the tumor. A total of $1 \times 10^6$ HER2.CARTs expressing *ffLuc* were systemically administered 3 days post injection of CAd*Trio*. Tumor volumes were monitored at different time points. **b** We collected serum samples from mice at 0, 3, 10, 24, 45, and 66 days post injection of CAd*Trio*, and measured human Th1 and Th2 cytokine levels by Multiplex. Data are presented as means ± SD. **c** Bioluminescence of HER2.CARTs was monitored at different time points. Data are presented as means ± SD, $p = 0.003$. *P*-values were determined using two-tailed *t* test (*t* ratio = 4.213, dF = 8). **d** CFPAC-1 tumors were harvested from non-humanized and humanized mice (non-humanized mice: $n = 4$ animals, humanized mice: $n = 5$ animals) at 3 days post injection of CAd*Trio*, and total RNA was extracted from whole tumors. Pro-inflammatory genes were quantified and normalized with human β-Actin. Data are presented as means ± SD. *P*-values were determined using two-tailed *t* test; $p = 0.0093$ (*t* ratio = 3.553, dF = 7), $p = 0.0004$ (*t* ratio = 6.352, dF = 7), $p = 0.0005$ (*t* ratio = 6.084, dF = 7), $p = 0.0045$ (*t* ratio = 4.101, dF = 7), $p = 0.014$ (*t* ratio = 3.226, dF = 7), $p = 0.027$ (*t* ratio = 2.782, dF = 7). Statistical significance set at $p < 0.05$, ns > 0.05. **e** CFPAC-1 tumors were harvested from humanized mice (non-humanized mice: $n = 4$ animals, humanized mice: $n = 5$ animals) at 3 days post injection of CAd*Trio*, and tumor infiltrating human immune cells were analyzed with flow cytometry. Box plot elements: central line, median; box limit, upper and lower quartiles; whisker, 1.5x inter-quartile range; points, outliers. Abbreviations: s.c. subcutaneous, i.t. intratumoral, i.v. intravenous, ND not detectable.

(Supplementary Fig. 15a). In this experiment, we additionally evaluated HER2.CART distribution and found initial HER2. CART infiltration and expansion in CAd*IL12*-treated mice was lower than that in mice treated with CAd*Trio* ($p < 0.01$) (Supplementary Fig. 15b), similar to our previous experiments with xenograft mouse models[14,15]. These data suggest that, while IL-12 may be the primary component of CAd*Trio* to induce HER2.CART infiltration and expansion, all CAd*Trio* components are required to maximize adoptively transferred HER2.CART infiltration and expansion at the primary tumor site and subsequent control of distant (CAd-untreated) tumors in humanized mice.

To compare how the treatments impact immune phenotypes, we euthanized all mice when control groups reached euthanasia criteria and phenotyped infiltrating immune cells in residual tumors (Fig. 5c, Supplementary Figs. 11, 12, and 13a). We found that mice treated with combination immunotherapy showed

higher endogenous human immune cell (CD45⁺, EGFP⁻) infiltration than other groups in their left (CAd*Trio*-untreated) tumors ($p = 0.0009$). However, there was no significant difference in the proportional composition of immune subsets (T cells, NK cells, monocytes, B cells) following combination immunotherapy compared to other groups. To address the correlation between immune cell infiltration and tumor immune gene signatures, we profiled the human immune gene expression signature of residual tumors (Fig. 5d, Supplementary Fig. 13b, Supplementary Data 1 and 2). Although there were no adenoviral vectors in the CAd*Trio*-untreated (left) tumors (Supplementary Fig. 14a), we found that type I IFN- and Th1-related genes were upregulated in the left tumors of mice treated with CAd*Trio* alone (but to a lesser degree than in right tumors), suggesting that local CAd*Trio* treatment stimulates systemic host immune responses against CFPAC-1 tumors. Since HER2.CARTs infiltrated both tumor sites in the absence of CAd*Trio* pre-treatment as well (Fig. 5b),

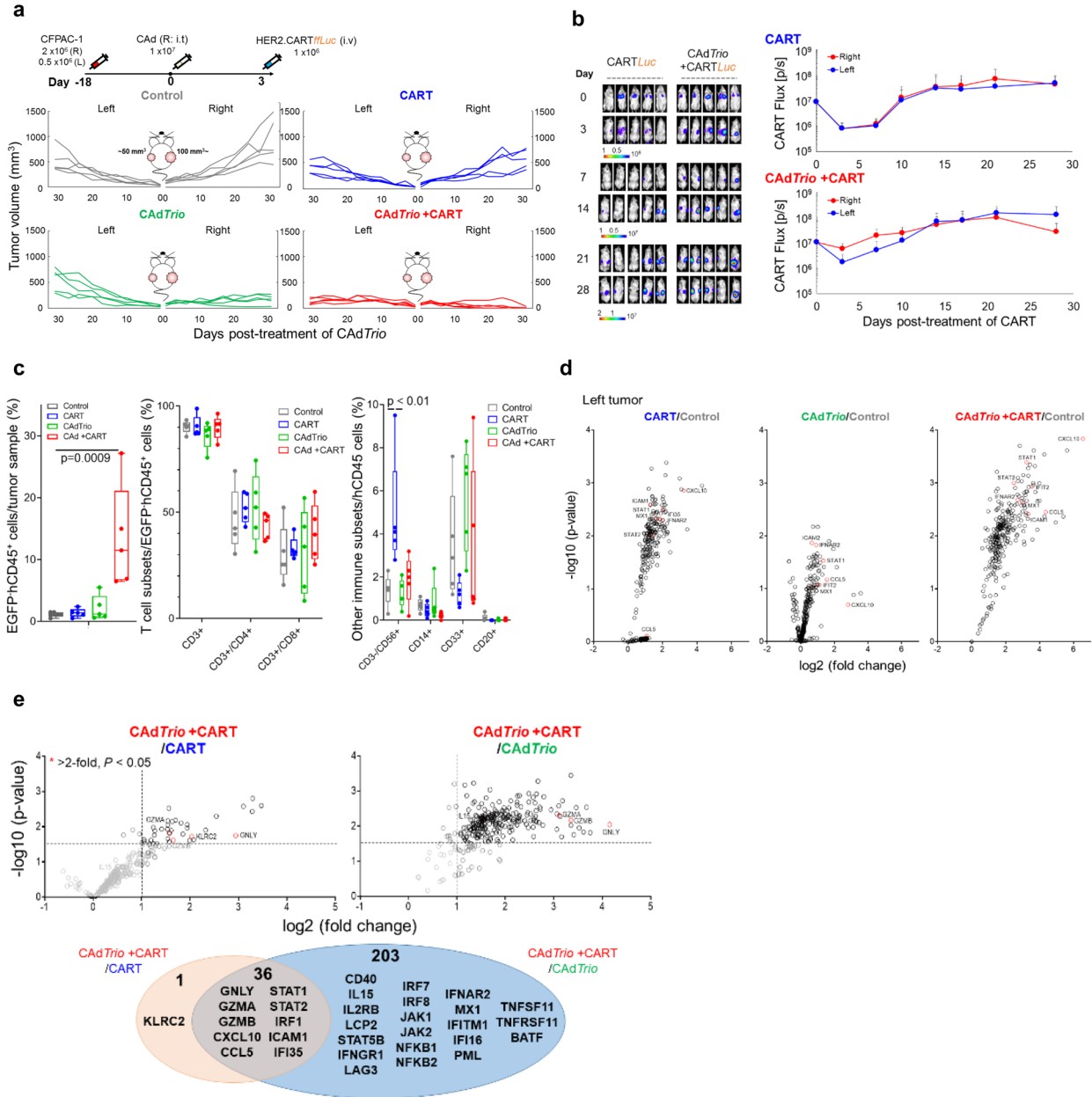

**Fig. 5 Combination immunotherapy controls tumor growth in humanized mice with multiple PDAC tumors. a** CFPAC-1 cells were transplanted into the right and left flanks of humanized mice ($n = 5$ animals). A total of $1 \times 10^7$ vp of CAd*Trio* (OAd:HD = 1:1) were injected into the right tumor. A total of $1 \times 10^6$ HER2.CARTs expressing *ffLuc* were systemically administered 3 days post injection of CAd*Trio*. Tumor volumes were monitored at different time points. **b** Bioluminescence of HER2.CARTs was monitored at different time points. Data are presented as means ± SD. **c** CFPAC-1 tumors were harvested from humanized mice at 31 days post injection of CAd*Trio*, and tumor infiltrating human immune cells were analyzed by flow cytometry. Box plot elements: central line, median; box limit, upper and lower quartiles; whisker, 1.5x inter-quartile range; points, outliers. *P*-values were determined using ordinary one-way ANOVA with Tukey multiple comparisons, $p = 0.0009$ (F(3,16) = 9.252), $p < 0.01$ (F(3,16) = 6.426). Statistical significance set at $p < 0.05$, ns > 0.05. Total RNA was extracted from whole tumor at 31 days post injection of CAd*Trio*. Gene expression was profiled with Nanostring, differential gene expression compared to control tumors (**d**) and compared to single agents (**e**) are shown. Abbreviations: s.c. subcutaneous, i.t. intratumoral, i.v. intravenous.

left tumors also expressed pro-inflammatory genes in the HER2. CART only condition (Fig. 5d). However, residual left tumors in mice treated with combination immunotherapy expressed higher levels of pro-inflammatory genes compared to mice treated with single therapies (Fig. 5d, Supplementary Data 1), and gene expression correlated with increased immune cell infiltration.

To address whether CAd*Trio* or HER2.CART primarily induced pro-inflammatory gene expression in residual tumors, we compared gene expression between combination and single

treatment profiles (Fig. 5e, Supplementary Data 3). We found that 239 genes were significantly upregulated in combination treatment compared to CAd*Trio* alone. Of these 239 genes, 36 genes upregulated in the combination group were associated with both CAd*Trio* and HER2.CART treatments. These 36 genes included a range of cytotoxic molecules such as Granulysin, Granzyme A and B. Only one gene (KLRC2 (NKG2C)) was uniquely upregulated in a CAd*Trio*-dependent manner (significantly upregulated in combination compared to HER2.CART alone).

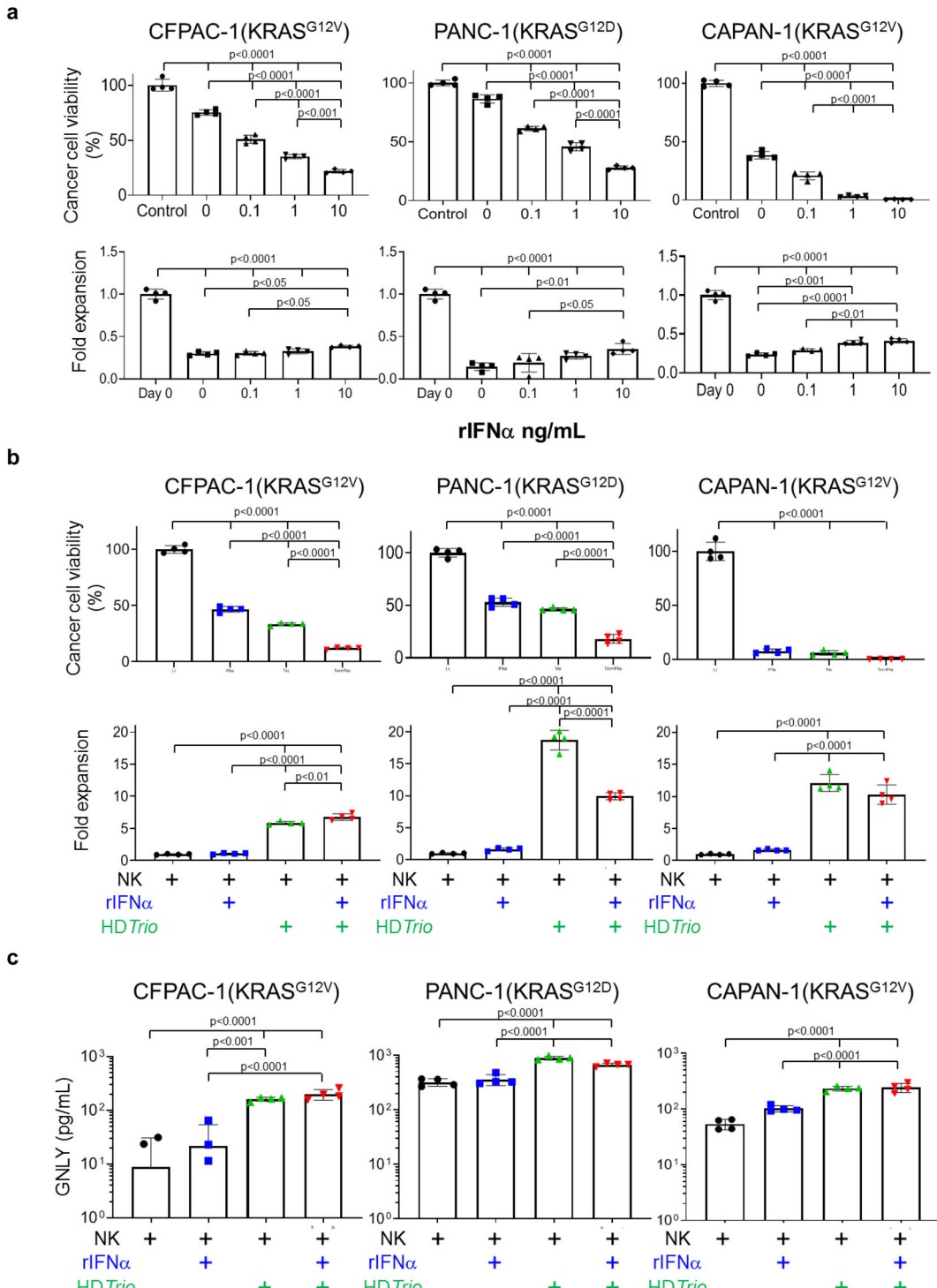

**NK cells enhance anti-PDAC activity in the presence of type I IFN and CAdTrio.** Although KLRC2 is primarily expressed in NK cells and effector CD8[+] T cells[34,35] we found KLRC2 upregulation 3 days after injection of CAdTrio and before injection of HER2.CARTs (Supplementary Fig. 16). Since lack of type I IFN signaling in NK cells impairs their cytotoxic function in different tumor models[36] and CAdTrio-treated tumors upregulated type I IFN, we hypothesized that type I IFN induced by immunomodulatory molecules derived from CAdTrio enhance the anti-PDAC activity of NK cells. We tested NK cell anti-PDAC activity using PDAC lines in the presence of recombinant type I IFN (rIFNα) in vitro (Fig. 6a). We found that rIFNα increased the

anti-PDAC activity of NK cells in a dose-dependent manner, yet having little effect on proliferation. Although rIFNα alone induced some cancer cell death (Supplementary Fig. 17), our results overall indicate that type I IFN enhances NK cell anti-PDAC activity. As IL-12 and PD-L1 blockade are also associated with improved NK cell anti-tumor activity[37,38], we next examined whether CAdTrio-derived IL-12 and PD-L1 blocker additively increased NK cell anti-PDAC activity (Fig. 6b). We found that these molecules enhanced NK cell anti-PDAC activity irrespective of the presence of IFNα, and independently induced NK cell proliferation. We next measured the cytotoxic molecule granulysin (GNLY) in media of these co-culture experiments, since

**Fig. 6 NK cells enhance anti-PDAC activity in the presence of type I IFN and immunomodulatory molecules from CAdTrio. a** PANC-1, CAPAN-1, and CFPAC-1 expressing *ffLuc* were cultured with NK cells with increasing doses of rIFNα. We also cultured NK cells expressing *ffLuc* with these cancer cells (E: T = 1:10) with increasing doses of rIFNα. Cells were harvested 0 and 72 h post coculture, and viable cells were analyzed by luciferase assay ($n = 4$ biologically independent samples, each timepoint). Data are presented as means ± SD. *P*-values were determined by ordinary one-way ANOVA with Tukey multiple comparisons. CFPAC ($F_{(4,15)} = 351.8$) NK ($F_{(4,15)} = 346.8$), Panc1 ($F_{(4,15)} = 468.7$) NK ($F_{(4,15)} = 106.3$), CAPAN1 ($F_{(4,15)} = 1121$) NK ($F_{(4,15)} = 318.4$. **b** PANC-1, CAPAN-1, and CFPAC-1 expressing *ffLuc* (or not expressing) were infected with 100 vp/cell of HDAdTrio. Cells expressing *ffLuc* were cultured with NK cells (E:T = 1:10). We also cultured NK cells expressing *ffLuc* with these cancer cells (E:T = 1:10) at 24 h post infection of HDAdTrio in the presence or absence of 1 ng/mL rIFNα. Cells were harvested 72 h post coculture, and viable cells were analyzed by luciferase assay ($n = 4$ biologically independent samples). Data are presented as means ± SD. *P*-values were determined by ordinary one-way ANOVA with Tukey multiple comparisons. CFPAC ($F_{(3,12)} = 1114$) NK ($F_{(3,12)} = 458.3$), Panc1 ($F_{(3,12)} = 356.6$) NK ($F_{(3,12)} = 410.2$), CAPAN1 ($F_{(3,12)} = 459.6$) NK ($F_{(3,12)} = 131.4$. **c** We sampled media 72 h post coculture, and measured granulysin using ELISA assay ($n = 4$ biologically independent samples). Data are presented as means ± SD. *P*-values were determined by ordinary one-way ANOVA with Tukey multiple comparisons, $p < 0.001$ ($F_{(3,12)} = 41.07$), $p < 0.0001$ ($F_{(3,12)} = 74.40$), $p < 0.0001$ ($F_{(3,12)} = 49.49$). Statistical significance set at $p < 0.05$, ns $> 0.05$.

GNLY can be induced by CAdTrio in humanized mice (Fig. 5e, Supplementary Fig. 16) and GNLY-expressing NK cells may be a prognostic marker of solid tumor responses to recombinant IFNα (rIFNα) therapy[34]. Although rIFNα slightly but not significantly increased GNLY levels in media compared to NK cells cocultured with PDAC alone, we found significantly increased levels ($p < 0.0001$) of GNLY in media in the presence of CAdTrio components (IL-12 and PD-L1 blocker), irrespective of rIFNα (Fig. 6c). These results indicate that GNLY expression from NK cells is primarily dependent on CAdTrio. Hence, type I IFN induced by CAdTrio treatment, combined with CAdTrio-expressed IL-12 and PD-L1 blocker, augments the activity and expansion of endogenous NK cells.

Overall, these studies indicate that CAdTrio treatment leads to additive host immune stimulation including NK cells against PDAC to aid adoptively transferred HER2.CART in the control of PDAC tumor growth.

## Discussion

Overall, we demonstrate that combining CAdTrio and HER2.CARTs is a curative immunotherapy in several PDAC models, including humanized mouse models.

One major limitation of CAR T cells for solid tumor treatment is the identification of cell membrane targets with high-level, homogenous expression on solid tumors and limited expression on normal cells[39]. CART can recognize antigens with relatively low expression due to higher overall avidity than antibodies, and our center has safely treated more than 36 patients with HER2-positive solid tumors with our HER2.CARTs[11,12]. A recent study showed that patient-derived xenograft (PDX) PDACs expressed targetable levels of HER2 by HER2.CARTs, and they also found that PDAC cancer-stem cells (CSCs) express similar levels of HER2 as the non-CSC population[40]. In this study, we found that our HER2.CARTs can recognize and kill PDAC lines expressing varying levels of HER2 in vitro and in vivo. However, we also found that tumors recurred in CAPAN-1 xenograft mice treated with HER2.CART alone due to antigen (HER2) loss, similar to what has been seen in CAPAN-1 xenograft mice treated with PSCA.CART[41]. These results suggest that CAPAN-1 attenuates the anti-tumor activity of single antigen-targeting CARTs with antigen loss independent of the target. Our combination treatment strategy quickly eradicates tumors overcoming the hurdle of antigen loss caused by protracted or incomplete tumor elimination by CAR T cells alone.

While CAR T-cell therapies for solid tumors have been effective in preclinical xenograft models, like our PDAC xenograft mice, several barriers to successful treatment exist in patients[39]. For instance, limited trafficking of cell therapies to tumor sites may hamper effective treatment of solid tumors[24,42]. We found

that delayed infiltration and expansion of HER2.CART at the tumor site attenuated anti-tumor activity in humanized mice. However, similar to another study using STING agonist[30], pretreating tumors with CAdTrio induced pro-inflammatory signals and chemotaxis, thereby improving CART infiltration within the tumor. Although we observed no significant difference in subsets of tumor-infiltrating lymphocytes between untreated control and CAdTrio-treated tumors in humanized mice, CAdTrio induced expression of type I IFN-related genes not detected in xenograft models. This finding suggests that IFN-related genes are activated in host immune cells, not tumor cells, and indicates that tumor-infiltrating immune cells recognize CAdTrio components through pattern recognition receptors (e.g., TLR9[29,31]) or are stimulated by CAdTrio transgenes to induce pro-inflammatory responses. Metzger et al. demonstrated that type I IFN stimulation through TLR agonist repolarizes the PDAC TME, specifically myeloid-derived suppressor cells, in an immunocompetent KPC model[43]. Although we did not see phenotypic differences with standard immune phenotype markers in this study, CAdTrio treatment and the resulting induction of type I IFN may repolarize tumor-infiltrated immune cells to an immune activated state.

Metastatic disease is another barrier to successful immunotherapy for PDAC patients. Local oncolytic viro-immunotherapy is intended to augment systemic host anti-tumor effects by inducing potent abscopal immune responses[44–46]. Given the inhibitory TME of human solid tumors, oncolytic virotherapy should be most effective against both injected and metastatic tumor sites when oncolysis is combined with relatively sustained immunostimulatory transgene expression[4]. Here, we used our CAd system to amplify the therapeutic transgenes encoded in HDAd with lytic effects provided through the OAd replication machinery[10]. Although the CAdTrio dosage used in this study showed limited anti-tumor effects in xenograft models, we found that this dose, with immunostimulatory molecules derived from HDAd, activated the host immune system in humanized mouse models resulting in significant tumor control. To address whether local CAdTrio treatment can systemically activate the host immune system and improve CART anti-tumor activity at a distant (untreated) site, we tested our combination immunotherapy in an advanced PDAC tumor model. Although HER2.CART primarily migrated to CAdTrio-treated tumor sites, they also expanded at distant tumors in mice treated with combination immunotherapy to significantly control tumor growth compared to single agent treatments. Treating humanized mice with CAd expressing single components revealed IL-12 as the component of CAdTrio most responsible for HER2.CART infiltration and expansion at the primary tumor site. However, none of the components alone eliminated the CAd-treated tumor or controlled distant tumor growth, as we observed in humanized mice treated with CAdTrio.

Our studies suggest that all components are required to control both primary and distant tumor growth. In future studies, we will use our humanized mouse models and samples from our ongoing Phase I clinical trial (NCT03740256) to further characterize which CAdTrio component(s) by which mechanism(s) modify the PDAC TME and influence HER2.CART anti-tumor activity.

In addition to stimulating adoptively transferred HER2.CART, in humanized mice CAdTrio induced active (Th1) gene signatures at distant (untreated) tumor sites (abscopal effect), similar to what has been seen in melanoma patients treated with T-VEC[47]. Therefore, CAdTrio treatment might make both tumor beds immunologically "hot" and thus susceptible to adoptively transferred HER2.CARTs. Combination treatment was required for significant control of distant tumors with relatively higher endogenous immune infiltration and type I IFN/Th1 related gene expression than other groups. However, single-cell RNA sequencing will be necessary to address which immune cells express these genes.

Based on immune gene signatures in residual tumors, we found that KLRC2 (NKG2C) was uniquely upregulated in a CAdTrio-dependent manner. KLRC2 is generally expressed on human NK cells and binds HLA-E molecules, indicators of adequate MHC class I expression by self cells[48]. Clinically, NKG2C+ NK cells are indicators of good prognosis in CMV- and HIV-infected patients and have been implicated in innate memory[34]. In cancer, increased infiltration of NK cells is associated with better prognosis in patients with solid tumors including PDAC[49,50]. However, attenuated type I IFN signaling by pancreatic cancer harboring KRAS mutation and expressing MYC oncogene limits NK cell activity at PDAC tumor sites[51]. Our in vitro data indicates that exogenous type I IFN can restore NK cell anti-PDAC activity independent of KRAS mutation and that CAdTrio components directly enhanced NK cell anti-PDAC activity and proliferation. These data suggest that CAdTrio exerts both indirect (via type I IFN induction by host immune cells) and direct (via IL12p70 and PDL1 blocker components) effects on host NK cells that contribute to anti-tumor immunity in our humanized models.

In summary, we demonstrate that local provision of multiple immunomodulatory molecules as a CAd package augments the anti-tumor effects of adoptively transferred HER2.CARTs and endogenous immune cells (e.g., NK cells), allowing them to control both CAdTrio-treated and -untreated PDAC tumors. Oncolytic viro-immunotherapy also induces antigen-spread, leading to the development of tumor-associated antigen-specific cytotoxic T lymphocytes in patients[4], an effect that cannot be fully replicated in our humanized mouse models due to limited thymic education/maturation[52]. Ultimately, clinical trials will reveal whether combination immunotherapy with CAdTrio and HER2.CART is sufficient for the desired systemic antitumor effects, including antigen spread. Analyses to address these questions are a major component of our Phase I clinical trial (NCT03740256) that has recruited its first patients following approval by the US Food and Drug Administration.

## Methods

**Adenoviral vectors (HDAds and OAds).** The IL-12p70, HSV thymidine kinase (HSVtk), and PD-L1 blocking mini-antibody (anti-PD-L1 scFv sequence obtained from Tessa Therapeutics Pte.) expression cassettes (IL-12p70 driven by EF1 promoter, HSVtk driven by hamster GRP78 promoter, and PD-L1 mini-antibody driven by human GRP94 promoter (InvivoGen)) were cloned into pHDΔ21E4 (HDAdTrio vector). After confirming the sequence and expression of IL12p70 (ELISA (BD Bioscience)), PD-L1 blocking mini-antibody (Western blotting with anti-HA IgG (Thermo Fisher Scientific)) and HSVtk (culturing cancer cells in the presence or absence of ganciclovir (GCV)), HDAdTrio were rescued with chimeric helper virus 5/3 (knob replacing Ad serotype 3 from Ad serotype 5) using 116 cells[53,54]. We measured infectious units (IUs) of HDAdTrio using A549 cells[53]. The IU of HDAdTrio used in this study was vp:IU = 100:11. HDAd without transgene

(HDAd0), HDAd expressing PD-L1 blocking antibody (HDAdPDL1), and HDAd expressing IL-12p70 (HDAdIL12) were cloned into pHDΔ28E4 vector and rescued in 116 cells[10,13,14]. We substituted and cloned chimeric OAd5/3Ad2E1AΔ24 (knob replacing Ad serotype 3 from Ad serotype 5) with Ad5E1AΔ24 to Ad2E1AΔ24 (deleting pRb binding site). We rescued OAd5/3Ad2E1AΔ24 using 293 cells and propagated using A549 cells[10]. All unique materials are available from the authors upon request.

**Cell lines.** We obtained human pancreatic lines CAPAN-1, CFPAC-1, and PANC-1 from ATCC (Manassas, VA) in 2018. We authenticated cell lines with Short Tandem Repeat (STR) profiling by ATCC. Cells were cultured under the conditions recommended.

To generate cell lines expressing the fusion protein EGFP-ffLuc, we infected cells with retrovirus encoding EGFP-ffLuc[13,14]. EGFP-positive cells were sorted using an SH800 Cell Sorter (Sony) after 3 passages post infection of retrovirus.

**Primary cells.** Human PBMCs were isolated using Ficoll-Paque Plus according to the manufacturer's instructions (Axis-Shield) from healthy donor whole blood (approved by the Baylor College of Medicine IRB Committee). The vector encoding the HER2-directed CAR incorporating the CD28 costimulatory endodomain (2nd generation HER2.28ζ.CAR)[21], the fusion protein EGFP-ffLuc, and the methodology for the production of retrovirus and CARTs have been described previously[11]. Briefly, PBMCs were activated with OKT3 (1 mg/ml) (Ortho Biotech) and CD28 antibodies (1 mg/mL) (Becton Dickinson) and fed every 2 days, beginning the day after stimulation, with media supplemented with 10 ng/mL of recombinant human interleukin-7 and 5 ng/mL of recombinant human interleukin-15 (rIL-7 and rIL-15, R&D). On day 2 post-OKT3/CD28 T blast generation, activated T cells (0.125 × 10⁶/mL) were added to CAR retroviral-coated plates and centrifuged at 400 × g for 5 min. CAR-transduced T cells (HER2.CARTs) were expanded with media supplemented with 10 ng/mL of rIL-7 and 5 ng/mL rIL-15. All unique materials are available from the authors upon request.

The methodology to expand human NK cells has been described previously[55]. Briefly, we co-cultured CD56+ PBMCs (enriched via positive magnetic column selection) with irradiated K562-mb15-41BBL at a 1:10 (NK cell:K562) in G-Rex cell culture devices (Wilson Wolf) in Stem Cell Growth Medium (CellGenix) supplemented with 500 IU/mL recombinant human interleukin-2 (NIH).

**Co-culture experiments.** FfLuc-expressing cancer cells were seeded in 48-well plates. We added either non-transduced T cells, HER2.CARTs, or NK cells 24 h later at the ratios described in Figure legends. We measured residual live cells (ffLuc activity) using a Luciferase Assay System (Promega) and measured by plate reader (BMG Labtech) at time points described in the Figure legends. We measured amounts of Granulysin using a granulysin ELISA kit (R&D).

**Flow cytometry.** We used the following fluorochrome-conjugated monoclonal antibodies: anti-human CD3, CD4, CD8, CD25, CD69, CD134, CD137, CCR7, CD45RO, PD-1, PD-L1, HER2, CD20, CD56, CD14, CD33, NKG2C, recombinant human HER2-Fc chimera, and anti-Fc (for detection of HER2.CAR) (BD Bioscience, Beckman Coulter, BioLegend, R&D systems). Cells were stained with these Abs or the appropriate isotype controls Abs for 30 min at 4 °C. We determined live/dead discrimination via exclusion of 7AAD positive cells (BD Pharmingen). Stained cells were analyzed using a Gallios flow cytometer (Beckman Coulter). We analyzed data with Kaluza software (BD Bioscience) according to the manufacturer's instructions.

**Animal experiments.** The Baylor College of Medicine Institutional Animal Care and Use Committee approved all animal experiments.

For the subcutaneous models, 2 × 10⁶ CFPAC-1 cells or 5 × 10⁶ CAPAN-1 cells were resuspended in a volume of 100 μL of PBS and injected into the right flank of 7–8 week-old NSG male or female mice. Fourteen days post transplantation, a total of 1 × 10⁷ vp of CAd (OAd:HDAd=1:1) were injected in a volume of 20 μL into the tumor. The ratio of OAd to HDAd in the CAd system was optimized to effectively propagate transgene(s) encoded in the co-injected HDAd with lytic effects even with clinically relevant dosages. Three days post injection of CAds, mice received 1 × 10⁶ HER2.CARTs intravenously. CARTs expressing ffluc were assessed using the In Vivo Imaging System (Xenogen)[13]. The endpoint was established at a tumor volume > 1500 mm³.

For the humanized mouse model, Newborn (1–2 day from birth) female and male NSGSGM3 (NSGTG^CMV-IL3,CSF2,KITLG Eav/mloySz: Jackson Laboratory) were sublethally irradiated (100 cGy) and intrahepatically injected with 5 × 10⁴ human cord-blood unit (CBU)-derived CD34+ cells. CBUs were obtained from MD Anderson Stem Cell Center, and CD34+ cells were isolated using CD34+ cell isolation kit (Miltenyi Biotech Inc.). After confirming human CD45+ cells in PBMCs of mice 8–9 weeks post injection, 2 × 10⁶ CFPAC-1 cells were resuspended in a volume of 100 μL of PBS and injected into the right flank of male and female humanized mice. Eighteen days post transplantation, a total of 1 × 10⁷ vp of CAd (OAd:HDAd=1:1) were injected in a volume of 20 μL into the tumor. Three days post injection of CAds, mice received 1 × 10⁶ autologous HER2.CARTs (same CBU derived PBMCs) intravenously. CARTs expressing ffluc were assessed using the In

*Vivo Imaging System* (Xenogen)[13]. The endpoint was established at a tumor volume > 1500 mm$^3$.

For humanized mice with two CFPAC-1 tumors per mouse, we injected $2 \times 10^6$ CFPAC-1 cells into the right flank and $1 \times 10^6$ CFPAC-1 cells into the left flank. Eighteen days post transplantation, we injected a total of $1 \times 10^7$ vp of CAd (OAd: HDAd=1:1) into the right flank tumor. Three days post injection of CAds, mice received $1 \times 10^6$ autologous HER2.CARTs intravenously. The endpoint was established at combined tumor volumes (right tumor volume plus left tumor volume) > 1500 mm$^3$.

**Isolation of tumor-infiltrating immune cells**. After rinsing harvested tumors with PBS, tumors were minced and incubated in RPMI media containing human tumor dissociation reagents (Miltenyi Biotec Inc.) at 37 °C for 1 h. Cells were passed through a 70-μm cell strainer (BD Pharmingen), and murine stroma cells were removed using a Mouse Cell Depletion kit (Miltenyi Biotech Inc.). Human cells were stained with the antibodies described in Results.

**RNA extraction and RT-PCR**. CFPAC-1 tumors were harvested from non-humanized and humanized mice 3 days post injection of CAd*Trio*, and RNA was extracted from whole tumors using RNAeasy Plus Mini kit (Qiagen). RNA samples were converted to cDNA using Super Script III First-Strand Synthesis System (Thermo Fisher). The levels of human pro-inflammatory cytokine/chemokine described in Results were quantified using CFX96 Real-Time PCR Detection System (Bio-Rad) and normalized with human β-Actin. We obtained all primer sets from Bio-Rad.

**Microarray analysis**. We harvested CFPAC-1 tumors from humanized mice 31 days post injection of CAd*Trio*. Total RNA was extracted from whole tumors using the RNeasy Plus Mini kit and quantified using the NanoDrop 2000 (Thermo Fisher Scientific Inc.). The Baylor Genomic & RNA Profiling Core performed RNA expression profiling with the nCounter Human Immunology V2 Panel (Nano-String Technologies). Data quality control, normalization, and advanced analysis were performed using nSolver 4.0 analysis software following NanoString analysis guidelines. Supplementary gene expression tables show expression level >2 and $p <$ 0.05.

**Quantification of vector genome DNA in Ad infected tumors**. We harvested PDAC tumors at time points described in the Figure legends. Total DNA was extracted from infected tumors, and vector copies were quantified with primer sets for OAd (5′- TCCGGTTTCTATGCCAAACCT-3′ and 5′- TCCTCCGGTGA-TAATGACAAGA-3′) and HDAd (5′-TCTGAATAATTTTGTGTTACTCA-TAGCGCG-3′ and 5′-CCCATAAGCTCCTTTTAACTTGTTAAAGTC-3′) and normalized with murine genomic GAPDH (5′- TAGGCCAGGATGTAAAGGT-CATTAAG-3′ and 5′- CCAGAAAGGTCACACGGCTAAA-3′)[10].

**Immunohistochemistry**. The Human Tissue Acquisition and Pathology Core at Baylor College of Medicine stained CFPAC-1 tumors from xenograft and humanized mice with anti-human CD45 antibody.

**Statistics and reproducibility**. Results are represented as means of twice or more independent experiments (biological replication). Data with three or more groups were analyzed by ordinary one-way ANOVA analysis. Wilcoxon matched pairs test was used to compare two groups of paired data. Data were analyzed with GraphPad Prism 9.

**Reporting summary**. Further information on research design is available in the Nature Research Reporting Summary linked to this article.

## Data availability

The Nanostring data are available on NCBI GEO (GEO accession number: GSE167113). Source data are available as Supplementary Data 4. All other data are available from the authors on reasonable request.

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

## Acknowledgements

The authors would like to thank Catherine Gillespie in the Center for Cell and Gene Therapy at Baylor College of Medicine for her editing of the paper. This work was supported by Tessa Therapeutics Pte. This work was also supported by NIH P01 CA094237 to M.K. Brenner, NIH P30-CA125123 to Human Tissue Acquisition and Pathology Core, NIH 5T32HL092332-17 to Dr. Helen Heslop and HHSH75R60219C00004 to MD Anderson Cord Blood Bank. Dysthe was supported by NIH 5T32GM088129-10. The content is solely the responsibility of the authors and does not necessarily represent the official views of the NIH.

## Author contributions

Conceptualization, M.S.; methodology, A.R., C.E.P., T.Y., W-C.M., M.K.M., Y.J., and M. D.; investigation, A.R., C.E.P., T.Y., W-C.M., M.K.M, Y.J., M.D., R.P., and M.S.; writing–original draft, A.R. and M.S.; writing–review & editing, M.S. and M.K.B.; supervision, M.S.; funding acquisition, M.K.B. and M.S.

## Competing interests

M.S. is a scientific consultant, and C.E.P is a consultant for Tessa Therapeutic Ltd. The other authors declare no competing interest.
