## [Peer Review File · Communications Biology]

Reviewers' comments:

Reviewer #1 (Remarks to the Author):

This is an interesting study utilizing the combination of replicating adenovirus encoding IL12 and anti-PDL1 in conjunction with HER2-targeted CAR-T. Pancreatic cancer is an interesting target for that modality given HER2 association with prognosis. However, the combination of an oncolytic virus with CAR-T is not a novel idea and has been explored by others using other OV's than the one described here. The results presented in the current study, similar to other studies I have seen are not overwhelmingly positive but do show possibility for additive effects and overcoming CAR-T resistance mechanism. The current study gives some correlative insight pointing to the role of innate immunity to this end but falls short of direct demonstration.

Below are listed my major questions and concerns:

The CAPAN-1 model results are interesting but given the complexity of the Ad component of the treatment, encoding IL12 and PDL1 antibody, it would be of high relevance to understand which component is the key component i.e the Ad, the IL-12, or PDL1. Is the combination absolutely necessary? The reason this is an important question is that the total treatment regimen is extremely complex and this may slow or prevent clinical translation. Understanding the minimal necessary components is key in this regard.

On page 9 (e.g. Fig 4d), what is the direct evidence that the type I IFN response is induced? CXCL10 is actually a type II IFN stimulated not type I. It would be warranted to look for type I IFN by some more direct measurement in order to make this claim or tone down this statement.

The results of the experiments in vitro experiments with NK cell activity against PDAC cell lines are not surprising. That said, it would have been very relevant to try to determine which is the dominant factor in the mouse model. For example, would an anti-IL12 antibody or anti IFN β change the response locally and at a distance? Again, having at least controls without IL12 and the PDL1 blocker in the Ad would help answer these questions. If the authors believe NK are responsible for the enhanced effect it would seem logical to also consider NK cell depletion experiments.

As a minor concern:

On page 6 the statement "attributable to extensive necrosis and disruption of tumor vasculature". There are other causes of resistance to virotherapy that are possible and well described, like IFN response for instance. Have the authors considered this in their models?

Reviewer #2 (Remarks to the Author):

The authors have studied the efficacy in pancreatic cancer of their previously described combination therapy approach in which HER2-specific CART cells are administered systemically while an intratumoral adenovirus cocktail is administered intratumorally comprising an oncolytic adenovirus mixed with a helper dependent adenovirus encoding IL12 and an anti-PDL1 antibody.

They originally demonstrated the power of the combination in head and neck cancer xenograft models and published the results in a very strong Molecular Therapy paper (2017).

The current manuscript which focuses on pancreatic cancer which, like head and neck cancers, is often HER2 positive. The main difference from the previously published study is that the authors have now evaluated the combination approach in the presence of a functional human immune system in immune-reconstituted tumor-bearing mice.

The findings are interesting and show that the intratumorally administered adenovirus cocktail can not only modulate the tumor microenvironment of the injected tumor, but can also "repolarize" the TME of a contralateral uninjected tumor, thereby improving the activity of the HER2 CART therapy.

The experiments are generally well designed and the immune cell infiltration data are convincing, even though the effect is not dramatic. Specific areas for attention are the following:

1. The statement in the abstract that "combining both agents (CART cells and adenoviruses) cured tumors in two PDAC xenograft models" is misleading. CART cells alone were curative in all five of the animals with CFPAC-1 tumors and in 3 of 5 animals with CAPAN-1 tumors. Adding the adenovirus served only to increase the number of animals cured from 3 to 5 of 5 animals in the CAPAN-1 group. This is not a significant finding.

2. Kaplan-Meyer survival plots have been omitted from Figures 4 and 5. These are particularly important data plots for the manuscript since they address the impact of the combination therapy in the CFPAC-1 model in immune-reconstituted mice which is the main novel aspect of the paper.

We thank the reviewers for their thoughtful feedback on our manuscript. We have incorporated their suggestions to the overall benefit of the paper. We have modified the **Abstract**, **Results** and **Discussion** sections in the highlighted text and added 3 new figures and revised 1 other in the **Results** section, including 1 main and 3 supplementary figures. Below are our responses to specific comments.

Reviewer #1:

-The CAPAN-1 model results are interesting but given the complexity of the Ad component of the treatment, encoding IL12 and PDL1 antibody, it would be of high relevance to understand which component is the key component i.e the Ad, the IL-12, or PDL1. Is the combination absolutely necessary? The reason this is an important question is that the total treatment regimen is extremely complex and this may slow or prevent clinical translation. Understanding the minimal necessary components is key in this regard.

We have indeed been optimizing individual components in our combinatorial approach and previously found that all components (oncolysis through OAd, blockade of the PD-1:PD-L1 interaction through PD-L1 blocking antibody, and cytokine signaling through IL-12) are necessary to maximize the anti-tumor effects of adoptively transferred CAR T-cells in multiple solid tumor xenograft mouse models, including PDAC (*Cancer Research* 2017, *Molecular Therapy* 2017, *Molecular Therapy* 2020). The FDA recently approved the use of CAAdTrio and HER2.CAR T-cells for patients with HER2 positive solid tumors and we have just recruited the first patients into this clinical trial at BCM (NCT03740256). Although our CAAd platform contains multiple components, the overall package is "off the shelf" and can be administered as a single injection to accompany and (hopefully) benefit CAR T-cells. We have extended the description of this information in the **Discussion** of our manuscript.

- On page 9 (e.g. Fig 4d), what is the direct evidence that the type I IFN response is induced? CXCL10 is actually a type II IFN stimulated not type I. It would be warranted to look for type I IFN by some more direct measurement in order to make this claim or tone down this statement.

We quantified type I IFN mRNA (IFN β) level in those samples and have now included these new RT-PCR data in **revised Figure 4d**. Humanized mice treated with CAAdTrio showed significant upregulation of IFN β mRNA ($p=0.0093$) compared to untreated controls. We also found significant upregulation of type I IFN downstream genes OAS1 ($p=0.0004$) and MX1 ($p=0.0005$) in these samples. While we agree that CXCL10 can be upregulated by type II IFN, previous reports demonstrated that CXCL10 can also be upregulated through type I IFN (*Molecular Therapy* 2013 doi: 10.1038/mt.2012.277, *Journal of Hematology & Oncology* 2019 doi: 10.1186/s13045-0190721-x, *Nature Communications* 2020 doi: 10.1038/s41467-020-17011-z, *Journal of Experimental Medicine* 2021 doi: 10.1084/jem.20200844). We now include these references in the revised manuscript, as well as our more compelling type I IFN data, and discuss this aspect of the work in the text.

-The results of the experiments in vitro experiments with NK cell activity against PDAC cell lines are not surprising. That said, it would have been very relevant to try to determine which is the dominant factor in the mouse model. For example, would an anti-IL12 antibody or anti IFN γ change the response locally and at a distance? Again, having at least controls without IL12 and the PDL1 blocker in the Ad would help answer these questions. If the authors believe NK are responsible for the enhanced effect it would seem logical to also consider NK cell depletion experiments.

We evaluated the anti-tumor effects of combination immunotherapy with CAd0 (no transgene), CAdIL12 or CAdPDL1 in humanized mice harboring two subcutaneous CFPAC-1 tumors. Anti-tumor activity and infiltration/expansion of adoptively transferred HER2.CART from these experiments are now included as **revised Supplemental Figure 14**. Although single CAd components (oncolysis, PD-L1 blocking antibody or IL-12) provide modest tumor control in both CAd-treated and untreated tumors in conjunction with HER2.CAR T-cells, only CAdTrio eliminates the CAd-treated tumors in humanized mice, so that all components are required to control primary (large) tumor growth in this model (**New Supplemental Fig. 14a**).

Additionally, in these experiments we evaluated HER2.CART distribution (**New Supplemental Fig. 14b**). These data suggest that IL-12 is necessary to induce infiltration and expansion of adoptively transferred HER2.CART at the primary tumor site. However, HER2.CART infiltration and expansion in CAdIL12-treated mice was significantly lower than in mice treated with CAdTrio at early time points ($p < 0.01$), similar to what we found in our previous studies with non-humanized mouse models (*Molecular Therapy* 2017, *Molecular Therapy* 2020). These results indicate that all CAdTrio components are required for HER2.CART infiltration and expansion at the primary (large) tumor site and subsequent tumor growth control of distant (CAd-untreated) site in humanized mice. We now include these results in both the **Results** and **Discussion** sections of the revised manuscript.

Although we did not see significant differences in tumor-infiltrating immune subsets by standard flow cytometry (**Fig. 5C**), we found that an NK cell related gene, NKG2C, was specifically upregulated by CAdTrio treatment in our Nanostring data (**Fig. 5E**). Based on these findings, we further investigated how NK cells contribute to CAdTrio-dependent anti-tumor effects in the current study. However, we also found upregulation of additional immune related genes through CAR T-cell treatment in conjunction with CAdTrio (**Fig. 5E**). We thus expect that endogenous NK cells are one of several contributory mechanisms by which our combination immunotherapy produces durable anti-tumor effects in humanized mice, and a more detailed assignment of the relative potency of each component will be the subject of future studies that will make use of material from our ongoing clinical trial referred to in response #1 above. We have clarified this point in the **Discussion**.

-On page 6 the statement “attributable to extensive necrosis and disruption of tumor vasculature”. There are other causes of resistance to virotherapy that are possible and well described, like IFN response for instance. Have the authors considered this in their models?

We agree that host anti-virus responses, including type I IFN, may suppress OAd replication as well as viral distribution, but the quoted statement refers to our immune-deficient xenograft mouse studies in this manuscript, which lack the anti-viral responses that could contribute to this effect. We previously found similar results in xenograft mice using other tumor models (*Molecular Therapy Oncolytics* 2014). Based on your comment, we have now clarified that this statement is made specifically in regard to xenograft mouse models. We hope to investigate how host immune responses to CAAd (like IFN responses) contribute to viral replication and persistence at the tumor site by using humanized mouse models and material from our ongoing clinical study.

Reviewer #2:

-The statement in the abstract that "combining both agents (CART cells and adenoviruses) cured tumors in two PDAC xenograft models" is misleading. CART cells alone were curative in all five of the animals with CFPAC-1 tumors and in 3 of 5 animals with CAPAN-1 tumors. Adding the adenovirus served only to increase the number of animals cured from 3 to 5 of 5 animals in the CAPAN-1 group. This is not a significant finding.

We agree and modified the **Abstract** in the revised manuscript.

-Kaplan-Meyer survival plots have been omitted from Figures 4 and 5. These are particularly important data plots for the manuscript since they address the impact of the combination therapy in the CFPAC-1 model in immune-reconstituted mice which is the main novel aspect of the paper.

We agree. We now include tumor volume-based Kaplan-Meyer plots for **Figure 4** in the revised manuscript as **Supplemental Figure 7**. As you can see, our immunotherapies (both single agents and combination) produced significantly better survival than untreated controls.

Although we showed how our immunotherapies impact long-term tumor control in the presence of endogenous immune cells/system in **Figure 4**, we did not address how they influence host immune cell infiltration at the tumor site because some tumors in mice treated with immunotherapies were eradicated or were too small to analyze. To address how these immunotherapies impact tumor-infiltrating endogenous immune cells, we decided to euthanize all mice at the same time point to compare tumor-infiltrating immune cells in **Figure 5** instead of following long-term animal survival. We include tumor volume-based Kaplan-Meyer plots for **Figure 5** in the revised manuscript as **Supplemental Figure 10**. However, there was no significant difference at this time point.

REVIEWERS' COMMENTS:

Reviewer #1 (Remarks to the Author):

The authors have sufficiently addressed my concerns.

We thank the reviewers for acceptance of our revised manuscript.

Reviewer #1:

The authors have sufficiently addressed my concerns.